# Microstructure-Based Relative Humidity in Cementitious System Due to Self-Desiccation

**DOI:** 10.3390/ma12081214

**Published:** 2019-04-13

**Authors:** Yong Zhang, Xiaowei Ouyang, Zhengxian Yang

**Affiliations:** 1Fujian Provincial University Research Center for Advanced Civil Engineering Materials, Fuzhou University, Fuzhou 350116, China; y.zhang-1@tudelft.nl; 2College of Civil Engineering, Fuzhou University, Fuzhou 350116, China; 3Microlab, Section of Materials and Environment, Department of 3MD, Faculty of Civil Engineering and Geosciences, Delft University of Technology, Stevinweg 1, 2628 CN Delft, The Netherlands; 4Guangzhou University-Tamkang University Joint Research Center for Engineering Structure Disaster Prevention and Control, Guangzhou University, Guangzhou 510006, China

**Keywords:** cementitious system, self-desiccation, relative humidity, microstructure, average pore diameter

## Abstract

The internal relative humidity (RH) plays a crucial role in most of the concrete properties. Self-desiccation caused by continuous cement hydration is a major factor affecting the RH of concrete. This paper investigates the relationship between RH and microstructure for cementitious systems in the case of self-desiccation. A series of paste specimens prepared with different binder and water-binder-ratio (w/b) were cured under sealed conditions from 1 day to 1.5 years. The RH and microstructure of the paste specimens were measured. The microstructure characteristics under study include porosity, pore size, evaporable and non-evaporable water content. The results reveal that the RH of cementitious system drops to a great extent in the first 105 days’ hydration and decreases slowly afterwards. The blended materials such as fly ash, slag or limestone powder have different influences on the RH. A mathematical model between RH and the average pore diameter is proposed for cementitious systems under self-desiccation, regardless of age, w/b or cement type.

## 1. Introduction

The internal relative humidity (RH) of cementitious system has been of great interest over recent decades due to its decisive role in cement hydration and potential impact on various engineering properties. Snyder and Bentz [1] observed that the cement hydration is suspended at 90% RH. By thermodynamic analysis, Flatt et al. [2] found that alite stops hydration at RH below 80% because of a negative capillary pressure that opposes the chemical reaction. The RH can significantly affect the gas permeation, the water absorption, the autogenous shrinkage, the chloride diffusion, etc. [3,4,5,6,7,8].

The drop of RH in a cementitious system can be attributed primarily to the continuous cement hydration, also referred to as self-desiccation. As the cement hydration proceeds, the free water in capillary pores is reacting and transforming into chemically bound water. The hydration products gradually fill the capillary pore space, reducing the RH where the pores remain saturated via the Kelvin-Laplace effect [1]. Meanwhile, the gas bubbles, i.e., air and water vapor, start to nucleate and grow in the larger pores [8]. Menisci are formed at the interface between the pore solution and water vapor. Upon the consumption of free water, the capillary pores become progressively smaller while the curvatures of the menisci become greater.

In cementitious systems, the RH level is determined as Equation (1) [9]:(1)$$\mathrm{RH}={\mathrm{RH}}_{S}\cdot {\mathrm{RH}}_{K}$$
where RH_S_ accounts for the water activity effect caused by dissolved ions. For an ideal solution, RH_S_ is estimated using Raoult’s law [10] and depends on the mole fraction of water present in the aqueous solution, as Equation (2).
(2)$${\mathrm{RH}}_{S}=\frac{n_{H_{2}O}}{n_{\mathrm{solution}}}$$

The term RH_K_ is associated with the curvature formed at fluid/vapour interfaces. If the adsorbed water film is taken into account, as illustrated in Figure 1, RH_K_ can be described with the Kelvin-Cohan Equation [11]:(3)$${\mathrm{RH}}_{K}=\exp\left(-\frac{2\gamma_{w}\cdot V_{m}\cdot {\cos\theta }_{w}}{(r_{p}-t)\cdot R\cdot T}\right)$$
where V_m_ is the molar volume of water; θ_w_ is the contact angle between water and solids (if perfect wetting is assumed, θ_w_ = 0); R is the ideal gas constant (8.314 J/mol∙K); T is the absolute temperature (K); γ_w_ is the surface tension of the fluid (0.072 N/m for pure water); r_p_ is the radius (m) of the pores in which the meniscus surface is formed; d_p_ (d_p_ = 2r_p_) is considered the smallest drained pore diameter [12]; t (m) is the thickness of the water film and depends on the RH level [12]:(4)$$t=\left[0.385-0.189\cdot \ln\left(-\ln\left(\mathrm{RH}\right)\right)\right]\times 10^{-9},\;[1\%\;\leq \;\mathrm{RH}\;\leq \;95\%]$$

By tracing the evolution of RH, one can capture the hydration kinetics of the cementitious system. The moisture distribution can also be determined, which influences the capillary pressure and water continuity in the cementitious system [4,12]. Accordingly, the autogenous shrinkage and transport properties can be predicted. In order to better understand and predict the engineering properties of cementitious materials, comprehensive knowledge of the time-related RH is essential. Changes in the RH caused by self-desiccation have been studied for many years in concrete literature. However, the relationship between microstructure and RH in the case of self-desiccation has rarely been addressed.

This paper investigates the microstructure characteristics and associated RH formed due to continuous self-desiccation in cementitious pastes. The paste specimens were prepared covering the factors of curing age (1 day to 1.5 years), water-binder-ratio (w/b = 0.4, 0.5 and 0.6) and supplementary cementitious materials (SCMs), i.e., fly ash, slag and limestone powder. Both RH and the microstructure of the paste specimens were measured. The obtained RH value in relation to microstructural parameters, i.e., porosity, pore size, non-evaporable water and evaporable water content, was analysed and discussed in depth.

## 2. Materials and Methods

### 2.1. Raw Materials

Cementitious paste specimens were cast. The raw materials for the binders were ordinary Portland cement (OPC), low calcium fly ash (FA), ground granulated blast furnace slag (BFS) and limestone powder (LP). The chemical composition and particle size distribution of the raw materials are given in Table 1 and Figure 2, respectively. On the basis of X-ray diffraction, the crystalline phase of FA was about 42 wt.%, compared to only 2 wt.% in BFS. The main content of the LP, CaCO_3_, was approximately 98 wt.%.

### 2.2. Sample Preparation

A range of paste mixtures were designed. The details of the mixtures for all binders are listed in Table 2. The binders were OPC, binary and ternary cements. The replacement levels of OPC by SCMs were 30% for FA, 70% for BFS and 5% for LP by mass of the binder. The water-binder-ratio (w/b) varied from 0.35 to 0.6.

Paste samples were prepared in 500 g batches. First, cement powders and deionized water were mixed at a low speed for 1 min and at high speed for 2 min. The fresh pastes were poured into plastic bottles and then vibrated continuously to remove air bubbles and sealed with lids thereafter. In order to avoid bleeding, the paste samples were rotated for one day before placing them in the curing room at 20 ± 0.1 °C. After periods of 1, 28, 105, 182, 370 and 575 days, the paste samples were taken out of the plastic bottles and crushed into small cubic pieces (around 1 cm^3^). Next, these pieces were categorized into two groups. The pieces in the first group were used for RH measurement [12]. In the second group, the pieces were immersed in liquid nitrogen at −195 °C for 5 min, and then placed in a freeze-dryer with a temperature of −24 °C and under vacuum at 0.1 Pa until the water loss was below 0.01% per day. One part of the dried samples was used for pore structure measurements and another part for thermogravimetric analysis (TGA). For TGA, the samples were further ground into powders (<63 µm) and then dried at 40 °C till a constant weight was achieved.

### 2.3. Measurement

#### 2.3.1. Measurement of Relative Humidity

The measurement of relative humidity RH was performed using Rotronic HygroClip2 sensors (Rotronic, Bassersdorf, Switzerland), which can measure the RH and temperature simultaneously. The nominal accuracy of the sensors is ±0.5% RH/±0.1 °C. Before and after each measurement, the sensors were calibrated with three saturated salt solutions at 65%, 80% and 95% RH. The temperature circumstance during the whole measurement was controlled at 20 ± 0.1 °C. The set-up for RH measurement is illustrated in Figure 3.

Cement paste pieces with desired ages from 1 day to 575 days were placed into a chamber and sealed thereafter. RH readings were logged at 2 min intervals using a data logger until stable RH data was derived. The specimen used for RH measurement must be as small as possible to easily reach the equilibrium of moisture. All RH values presented in this paper were derived based on the average reading of two sensors. In all cases, the absolute differences between the two sensors were 1% at most. The differences can be ascribed to the systematic and random errors during RH calibration and measurements.

#### 2.3.2. Measurement of Pore Structure

Mercury intrusion porosimetry (MIP) is a technique commonly used for investigating the pore structure characteristics of cement pastes. Its principle is to immerse a specimen with a non-wetting liquid, viz. mercury, and force the liquid by applying increasing pressure. In general, the intruded pores are assumed to be cylindrical in shape. According to the Washburn Equation, the pore radius r_p_ (μm) is inversely proportional to the applied pressure P (MPa):(5)$$r_{p}=\left(-2\gamma_{\mathrm{Hg}}\cos\theta \right)/P$$
where γ_Hg_ stands for the surface tension of mercury (0.48 kN/m) and θ is the contact angle between solid and mercury (139°).

Although MIP has been widely recognised as a measure to test pore structure parameters, this technique does impose drawbacks. MIP misrepresents pore sizes because it measures pore size on the basis of the diameter of accessible throat pores through which the mercury penetrates the microstructure to reach internal pores. This inherent error in MIP is called the “ink-bottle effect” [13], which results in an overestimation of small capillary pores and meanwhile, an underestimation of large capillary pores.

In this study, the pore size distribution of cement paste was measured using a Micromeritics PoreSizer 9320 (Micromeritics, Norcross, GA, USA). Each measurement was conducted in three stages: (1) a manual low-pressure intrusion run from 0 to 0.17 MPa; (2) an automated high-pressure intrusion run from 0.17 to 210 MPa; (3) an automated high-pressure extrusion run from 210 MPa down to 0.03 MPa. The equilibrium time for each level of applied pressure was controlled at 30 s. A detailed description of the test procedure can be found elsewhere in [14]. For each material, three parallel measurements were performed and the average value was used for analysis. A typical representation of the MIP-derived pore size distribution is provided in Figure 4.

The pore structure characteristics were evaluated in two ways. The total porosity, noted as φ, was determined according to the cumulative intrusion volume at the maximum pressure of 210 MPa. Representing the total volume V_t_ and total surface S_t_ of capillary pores, the average pore diameter d_a_ can be determined using Equation (6).
(6)$$d_{a}=\frac{4V_{t}}{S_{t}}$$

#### 2.3.3. Thermogravimetric Analysis

Thermogravimetric analysis (TGA) was used to measure the mass loss due to ignition. The TGA was performed using an STA (TG-DTA-DSC) 449 F3 Jupiter (NETZSCH, Selb, Germany). Samples of about 50 mg were placed in an aluminum oxide crucible. Prior to the analysis, the samples were dried under N_2_-purging in the equipment at 40 °C to avoid carbonation while monitoring their mass loss. The dried samples were then heated from 40 to 1000 °C with a heating rate of 10 °C/min while the oven was purged with N_2_ at 50 mL/min.

The weight loss during heating can be attributed to the water release from different hydrates and possibly some adsorbed water or other gas (e.g., CO_2_). In the range of 400–550 °C, a sharp weight loss step was observed due to the decomposition of calcium hydroxide (CH). At temperatures above 700 °C, calcium carbonate (CC) was decomposed and the associated weight loss was registered as CO_2_. Since other hydrates, e.g., C–S–H, can decompose in the same temperature interval, the weight loss due to CH or CC was determined using the tangential method [15]. The non-evaporable water content, W_n_, is expressed in Equations (7) and (8).
(7)$$W_{n}=\frac{w_{1}-w_{2}}{w_{2}}-\frac{I_{\mathrm{bc}}}{1-I_{\mathrm{bc}}}$$
(8)$$I_{\mathrm{bc}}=p_{b}\cdot I_{b}+p_{c}\cdot I_{c}$$
where w_1_ and w_2_ are the weight of specimen before and after ignition, respectively. w_1_ is the specimen weight at 40 °C rather than 100 °C, because ettringite (AFt) and C–S–H have already decomposed at 100 °C [16,17]. w_2_ is the weight of the specimen dried at 1000 °C. p_b_ and p_c_ are the weight percentage of blended material and OPC, respectively. I_b_ and I_c_ are the loss on ignition of blended material and OPC, respectively.

The evaporable water content is expressed as the volume ratio of void space occupied with evaporable water (dried at 105 °C) in 1 g cement paste. The volume of evaporable water is equal to the difference between the initial total water content before casting and non-evaporable water content after hydration, with both parameters normalized to the volume of cement paste. For 1 g cement paste with a w/b of 0.5, the volume of voids V_v_ and evaporable water V_w_ can be expressed as:(9)$$V_{v}=\frac{1}{\rho_{b}}\cdot \varphi$$
(10)$$V_{w}=\frac{0.5}{1+0.5}-\frac{1}{1+0.5}\cdot \left(1-I_{\mathrm{bc}}\right)\cdot W_{n}$$
where ρ_b_ and φ represent the bulk density and total porosity of cement paste, respectively. I_bc_ is the loss on ignition of blended cement, see Equation (8).

#### 2.3.4. X-ray Diffraction

X-ray diffraction (XRD) was used mainly to study the mineralogy of the cementitious systems with and without LP. For mineralogical investigations, the powders were first mixed with standard crystal Al_2_O_3_ powders and then pulverized to an average particle size of less than 10 μm with no particle feeling. After that, about 3 g of the powder specimens was measured with XRD using a PANalytical X’Pert Pro MPD diffractometer (PANalytical B.V., Almelo, The Netherlands) with an incident beam monochromator and CuKα radiation (λ = 1.54 Å). The measurement was operated at 40 kV, 30 mA. The 2θ angle was from 5° to 70° with a step size of 0.03°. The Rietveld analysis was performed to quantify the crystalline components by comparing with the standards established by the International Centre for Diffraction Data.

## 3. Results and Discussion

The experimental results are organized into four groups. The development of RH values due to self-desiccation in paste specimens are presented first. The subsequent three groups describe the microstructure characteristics and their correlations to the RH of the paste specimens, including porosity vs. RH, average pore size vs. RH, and evaporable water content vs. RH.

### 3.1. Relative Humidity RH of Cementitious Pastes

RH measurements were performed in duplicate pairs and the average value was used in the plots. In all cases, the differences between duplicate measurements were within ±1%. For hardened cement pastes with sealed curing, the RH exhibits a homogeneous distribution due to no ingress of extra water. The free water in the capillary pores will be gradually consumed with cement hydration and transformed into bound water, filling the capillary pore space. As a result, the RH is expected to decrease with cement hydration, known as self-desiccation.

#### 3.1.1. RH in OPC Cement Paste

Figure 5 plots the development of RH due to self-desiccation up to 575 days in the OPC pastes with a water-cement-ratio (w/c) from 0.35 to 0.6. The data clearly indicate that the RH value significantly decreases in the first 105 days, followed by a gradual decrease afterwards. The phenomenon of self-desiccation is substantially enhanced with the decrease of w/c. These findings are in good agreement with those from previous studies [18]. Based on the data shown in Figure 5, the RH is negatively and exponentially correlated to the curing age (t), as expressed by Equation (11):(11)$$\mathrm{RH}\left(t\right)=A\cdot e^{-\left(\frac{t}{n}\right)}+{\mathrm{RH}}_{0}$$

There are three parameters in the above equation, A, n and RH_0_. Parameter A is related to the material property and mainly affected by w/c. n denotes the ageing effect due to continuous cement hydration. RH_0_ represents the humidity level at infinite curing age. These parameters and correlation coefficient R^2^ corresponding to different w/c are tabulated in Table 3.

For further analysis, the effect of w/c on the RH value at different ages is illustrated in Figure 6. As can be seen, the RH is expressed as a function of w/c. For mature cementitious systems, i.e., after 182 days’ hydration, the evolution of RH is limited and the curves thereafter appear to overlap. Logarithmic relationships can be derived, regardless of the age. Take the RH data at 182 days for example, the RH is logarithmically described as Equation (12) with a near-perfect correlation coefficient of 0.989.
(12)$${\mathrm{RH}}_{w/c}=0.233\cdot \ln\left(\frac{w}{c}\right)+1$$

By combining Equation (11) and Equation (12), the RH of OPC pastes can be determined as a function of age t and w/c, as indicated in Equation (13). This equation serves as a tool to predict the RH value due to long term self-desiccation in OPC pastes. It provides a reference for the initial mixture design as well.
(13)$$\mathrm{RH}={\mathrm{RH}}_{t}\cdot {\mathrm{RH}}_{w/c}=\left[A\cdot e^{-\left(\frac{t}{n}\right)}+{\mathrm{RH}}_{0}\right]\cdot \left[0.233\cdot \ln\left(\frac{w}{c}\right)+1\right]$$

#### 3.1.2. Effect of SCMs on RH

There is no doubt that the partial replacement of Portland cement by SCMs, either reactive or inert, will affect the RH because of the changes in the chemical constitutions and changes in the particle size distributions, as well as the changes in the hydration process and related changes of the microstructure formation in hardened cementitious systems.

Figure 7 presents the time-related RH in the paste specimens made with different SCMs. In order for a comparison to be made, the data of neat OPC paste P50, as a reference, is displayed as well. Regardless of the w/b, the plots for slag-blended pastes (PB40, PB50, PB60 and PBL50) are all below those of the reference P50. This indicates that the addition of slag can significantly reduce the RH in cementitious systems. On the contrary, the plots for FA-blended pastes are a little higher than those of the reference P50 in the entire curing age. It is of great interest to note that the w/b plays a less important role in the RH for slag-blended pastes (PB40, PB50 and PB60) than for OPC pastes (P40, P50 and P60). At 182 days, the RH of P60 (w/c = 0.6) is 8.46% higher than that of P40 (w/c = 0.4), see Figure 5, compared to a RH difference of 4.4% between PB60 and PB40, see Figure 7.

Replacing OPC by FA or slag increases the effective w/c and hence increases the relative amount of free water in the capillary pores. The ingredient SiO_2_ in FA/slag can react with Ca(OH)_2_ and produces the secondary C–S–H, filling the capillary pores. Compared to the FA, the slag has a much stronger pozzolanic reactivity and thus consumes more capillary free water and produces a finer pore structure. Moreover, due to the latent hydraulic reactions [19], the slag not only reacts with CH, but also consumes water at later ages. All these support that the slag has a higher capability than FA in reducing the RH.

With further addition of LP, the paste PFL50 has a slightly lower RH than the paste PF50. Whereas, the RH of the paste PBL50 is much higher than that of the paste without LP, i.e., PB50. It has been reported that LP addition can influence the chemical constitution of the alumina-contained phases (AFm) in cementitious systems [20]. A very small amount of CO_3_^2−^ content could lead to rapid consumption and decrease the amount of monosulfoaluminate [21]. At 22 °C and pH > 12, the SO_4_^2−^ groups in the interlayer region of monosulfoaluminate are easily substituted by CO_3_^2−^; therefore, the crystallization of hemicarboaluminate (Hc) or monocarboaluminate (Mc) takes place. According to Kuzel [22]:
C_3_A + 1/2CH +1/2CC+ 11.5H_2_O → C_3_A·1/2CC·1/2CH·11.5H_2_O (Hc)(14)
C_3_A + CC + 11H_2_O → C_3_A·CC·11H_2_O (Mc)(15)

The combination of FA and LP promotes the formation of alumina-carbonate compounds and the transformation of ettringite to monosulphate at a later hydration period is hampered because the presence of alumina-rich FA lowers the sulphate to alumina ratio. The stabilisation of water-rich bulky ettringite, instead of producing monosulphate, results in a decrease in the total free water in the capillary pores and an increase in the total volume of solid hydrates [16,21]. The addition of 5 wt.% LP can at first dilute the cementitious system, but the dilution effect is counteracted by the stabilization of the bulky ettringite as well as by the chemical formation of alumina-carbonate compounds. As a result, the RH of paste PFL50 is similar to (i.e., slightly lower than) that of the paste PF50.

Rietveld analysis was used to quantify the amount of ettringite, Mc and CH relative to the dry cement content. The results are given in Table 4. The paste specimens were cured for 182 days. Much more ettringite is formed in the binary system PF50 than in PB50. With further addition of 5 wt.% LP, the ternary system PFL50 presents even more ettringite content, and a considerable amount of Mc is produced in PFL50 as well. In contrast, for BFS-blended systems, the content of ettringite seems unchanged with and without LP when comparing PBL50 to PB50; however, more Mc is formed with the addition of LP.

With 5 wt.% addition of LP, more products (i.e., Ettringite, Mc) are formed in system PFL50 than in system PBL50. This proves that there is a synergistic effect between LP and FA in filling the capillary pores, which can be attributed to the high alumina content in the raw FA.

### 3.2. Total Porosity and RH

Figure 8 displays the total porosity as a function of age for all paste mixtures. The bold solid lines emphasize the slag-blended pastes, the thinner solid lines indicate the FA-blended pastes and the grey dotted lines represent the reference neat OPC pastes.

The general trend of total porosity is declining with age. The most significant decrease occurs in the first 105 days. Afterwards, the total porosity decreases in a gradual manner. With the progress of cement hydration and pozzolanic reaction of SCMs, more hydration products, e.g., C–S–H, are produced and precipitated in the capillary pores, resulting in a lower total porosity. The total porosity is remarkably decreased when the w/b changes from 0.4 to 0.6, regardless of OPC or slag-blended cement.

Compared to the reference OPC paste P50, the inclusion of FA (PF50) increases the total porosity at all ages. The higher total porosity in FA-blended paste can be ascribed to the fact that a relatively higher w/c ratio (lower gel to space ratio) is created when cement is partially replaced by FA [26,27]. However, opposite observations are obtained with the inclusion of slag. At a given w/b ratio, the total porosity in slag-blended paste is much lower than that in OPC paste. This can be ascribed to the lower density of slag powders, leading to a higher solid to void ratio by volume at the beginning of casting. The more important reason is the occurrence of strong pozzolanic and latent hydraulic reactions in the slag-blended cementitious systems [19]. The resultants, C–S–H, densify the microstructure by transforming coarse pores into finer ones. As a result of a much greater amount of amorphous phase the slag shows much higher reactivity than the FA.

The addition of LP is found to increase the total porosity when comparing the total porosity between PBL50 and PB50 and PFL50 and PF50. It is noteworthy that the porosity increment due to LP addition is much lower in the FA-blended system than in the slag-blended system. For instance, at 182 days, the total porosity is 7.79% higher in PBL50 paste than in PB50 paste, while the total porosity of PFL50 paste is only 2.52% higher than that of PF50 paste. The relatively lower porosity increment in PFL50 results from the strong synergistic effect between the alumina-rich FA and the LP, resulting in the stabilization of the bulky ettringite [28].

Figure 9 plots the RH against the total porosity for binders with and without SCMs. As can be seen, there are two sets of data corresponding to OPC paste specimens and blended paste specimens. Linear relationships between RH and total porosity are found in both OPC pastes and blended pastes. However, the linear curve of the blended pastes is clearly below that of the OPC pastes. This observation manifests that with equal total porosity, the blended pastes show lower RH compared to the OPC pastes. This, in turn, indicates that much smaller pores are present with the inclusion of SCMs. The dominant effect of pore size on the RH will be discussed in detail next.

### 3.3. Average Pore Diameter and RH

Figure 10 shows the relationship between the RH and average pore diameter for various cementitious mixtures. In this study, the pore solutions for all binders at 28 days were extracted and performed with inductively coupled plasma optical emission spectrometry (ICP-OES) measurements. The amount of cation in the pore solution of paste specimens at 28 days, as a representative, is provided in Table 5. The content of the anion (e.g., OH) is considered almost equal to the combined content of the cation (e.g., Na and K) [29]. The RH_S_ was calculated according to Equation (2). Upon obtaining the measured RH, the RH_k_ can then be determined by using Equation (1). The average pore diameter was calculated using Equation (6) based on the cumulative intrusion volume derived from MIP. The data set shown in Figure 10 proves an inherent correlation between the RH and average pore diameter in cementitious systems under self-desiccation. The best fit for the experimental data can be described using an exponential relationship, given in Equation (16).
(16)$${\mathrm{RH}}_{k}=e^{-\frac{1.23}{d_{a}-5.7}},{\;R}^{2}=0.953$$
where RH_k_ indicates the relative humidity controlled by the curvature effect, d_a_ denotes the average pore diameter. Herein, the d_a_-value is assumed to be higher than 5.7 nm, which is reasonable in cementitious materials [14].

The fitting curve apparently shows an intimate relationship between RH_k_ and average pore diameter d_a_ in cementitious systems cured under sealed conditions. In comparison to the simulated curve via Kelvin’s law, the fitting curve tends to slightly underestimate the average pore diameter. The main reason is attributable to the inherent ink-bottle effect of MIP measurement, which leads to an underestimation of large pores and an overestimation of small pores. The high correlation coefficient, R^2^ = 0.953, emphasizes that, in case of self-desiccation, the RH is dominated by the average pore diameter in the cementitious systems, regardless of age, cement type or w/b ratio. On this basis, it is reasonable to consider that the smallest drained pore diameter d_p_ equals the average pore diameter d_a_ in the cementitious material under self-desiccation.

### 3.4. Non-Evaporable Water/Evaporable Water and RH

Figure 11 shows the development of non-evaporable water content Wn normalized to the mass of paste specimens dried at 1000 °C. The specimens were cured from 28 days up to 370 days, with a constant w/b of 0.5. It can be seen that the Wn in blended pastes is lower than that in plain OPC paste at the same age. A slightly higher content of Wn in PB50 than in PF50 after 105 days’ hydration confirms a relatively higher chemical reactivity of slag than FA. Whereas, the Wn at 28 days is higher in PF50 than in PB50. The main reason is that in PF50 only 30% of OPC is replaced by FA, compared to a high replacement level of 70% by slag in PB50. The inclusion of LP (5 wt.%) decreases the non-evaporable water content Wn when comparing the ternary systems (PFL50 and PBL50) and their corresponding binary systems (PF50 and PB50). The capability to consume water shows a descending order as: OPC > BFS > FA > LP.

With a minor amount of LP addition, i.e., 5 wt.%, the FA-blended system exhibits a smaller reduction in Wn compared to the slag-blended system. For instance, at the age of 105 days, the Wn is 2.26% lower in PFL50 than in PF50, compared to 5.46% lower in PBL50 than in PB50. This indicates that LP addition consumes relatively more water in the FA-blended system than in the slag-blended system, which provides further evidence of the stabilization of water-rich ettringite in the FA-blended system.

Figure 12 plots the relationship between evaporable water content and RH in five different paste mixtures. Compared to plain OPC paste, the blended pastes present higher evaporable water content corresponding to lower RH. This can be ascribed to the finer pore size distribution in the blended pastes. The finding is consistent with the conclusion presented in the Section 3.3. There seems no inherent correlation between RH and evaporable water content, given that the two parameters cannot be fixed in view of different binders.

## 4. Conclusions

The development of RH in various cementitious pastes cured under sealed conditions, i.e., self-desiccation, was investigated. The roles of cement type, w/b ratio and age on the RH value were examined by studying their effects on the microstructure characteristics including porosity, average pore size, non-evaporable water and evaporable water content. The main findings are summarized as follows.

■In the case of self-desiccation, the RH declines with age. A significant decrease occurs in the first 105 days, followed by a gradual decrease.■The inclusion of slag decreases the RH, while FA or LP addition increases the RH. In the FA-OPC system, further addition of 5 wt.% LP slightly decreases the RH. Whereas in the slag-OPC system, further addition of LP 5 wt.% greatly increases the RH. A combination of alumina-rich FA and LP shows a synergistic effect, which results in the formation of carboaluminate and stabilization of water-rich bulky ettringite.■The pastes blended with FA or slag exhibit lower RH than the OPC pastes of a given total porosity.■There is an inherent relationship between RH and the average pore diameter in cementitious systems, regardless of cement type, w/b ratio or age.

## Figures and Tables

**Figure 1 materials-12-01214-f001:**
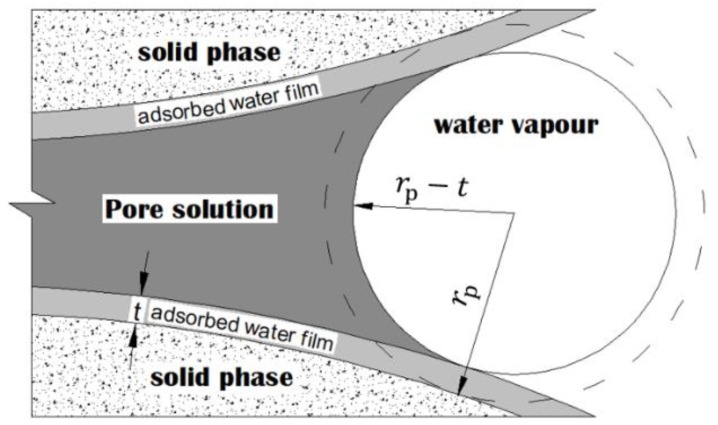
Schematic illustrations of meniscus curvature and adsorbed water film in capillary pores.

**Figure 2 materials-12-01214-f002:**
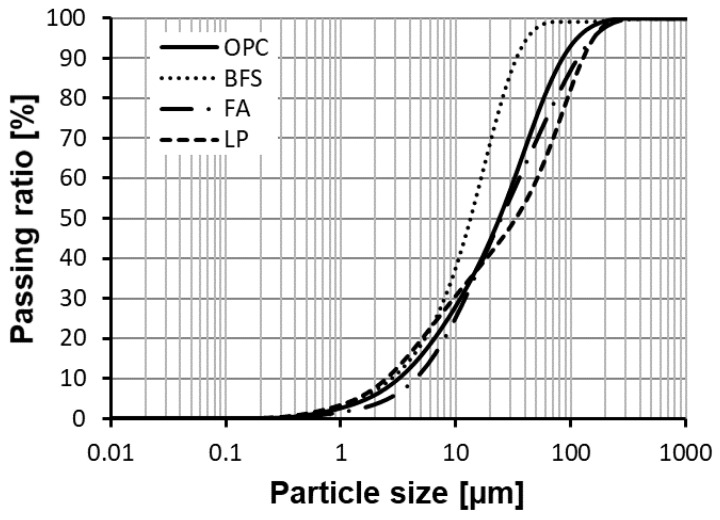
Particle size distribution of various powders by laser diffractometry.

**Figure 3 materials-12-01214-f003:**
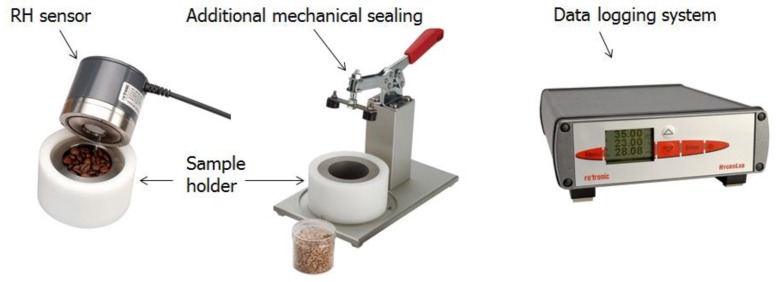
Set-up for the RH measurement.

**Figure 4 materials-12-01214-f004:**
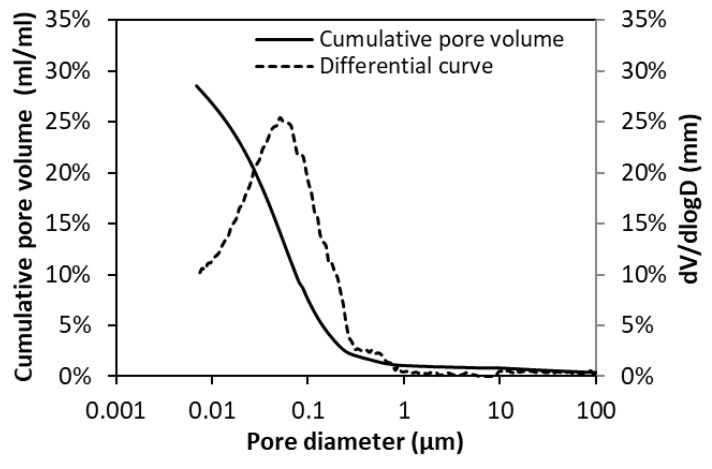
Pore size distribution of cement paste (OPC, 28 d, w/b = 0.5) measured by mercury intrusion porosimetry (MIP).

**Figure 5 materials-12-01214-f005:**
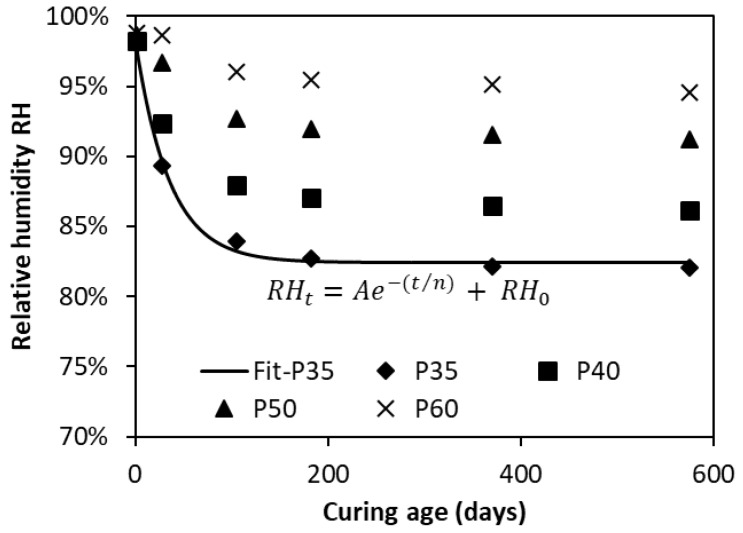
Development of relative humidity with cement hydration in OPC pastes.

**Figure 6 materials-12-01214-f006:**
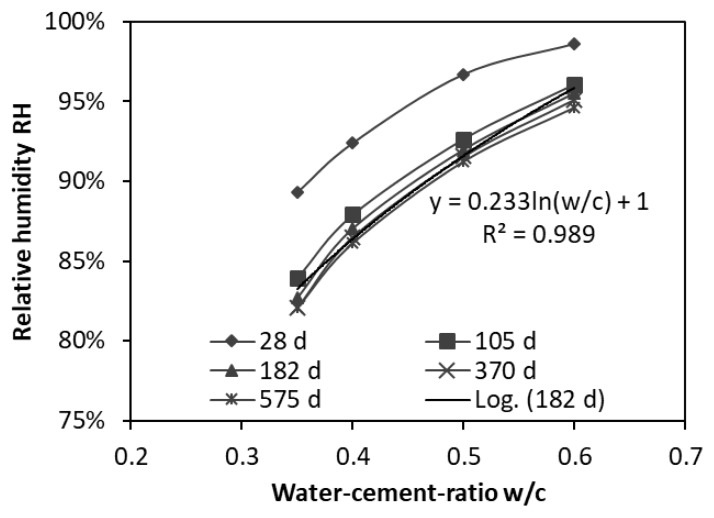
Effect of water-cement-ratio on time-related RH in OPC pastes.

**Figure 7 materials-12-01214-f007:**
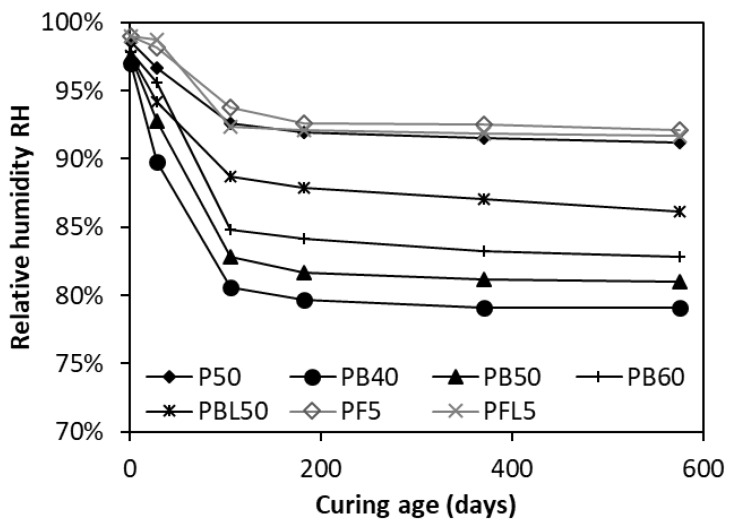
Effect of SCMs on relative humidity in blended pastes.

**Figure 8 materials-12-01214-f008:**
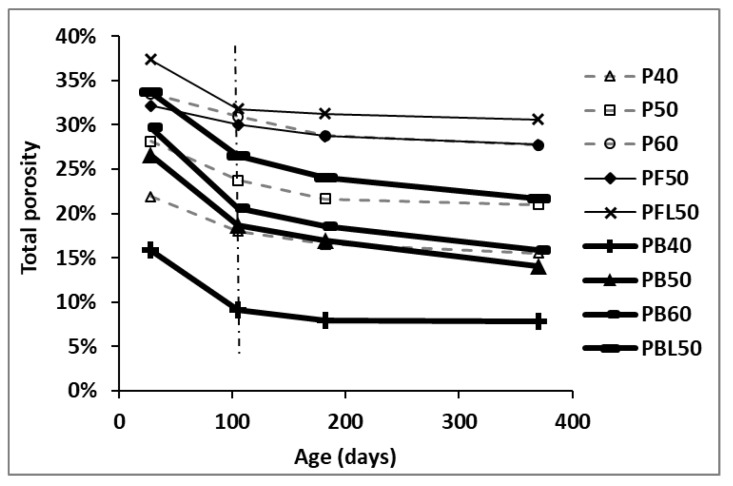
Evolution of total porosity with age in various cementitious pastes.

**Figure 9 materials-12-01214-f009:**
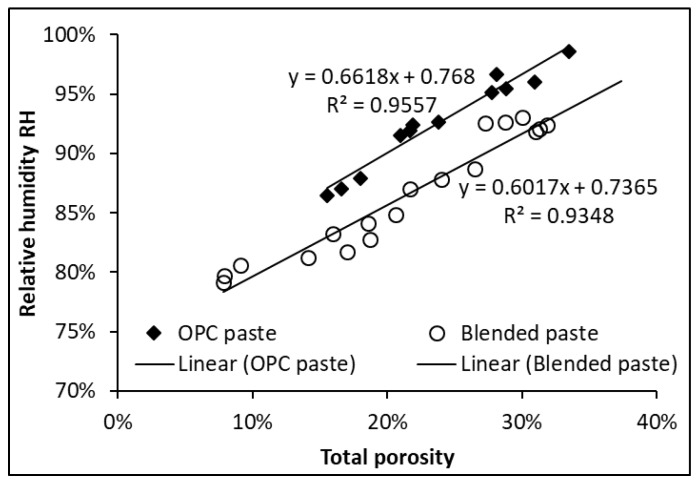
Relative humidity and total porosity in OPC and blended pastes.

**Figure 10 materials-12-01214-f010:**
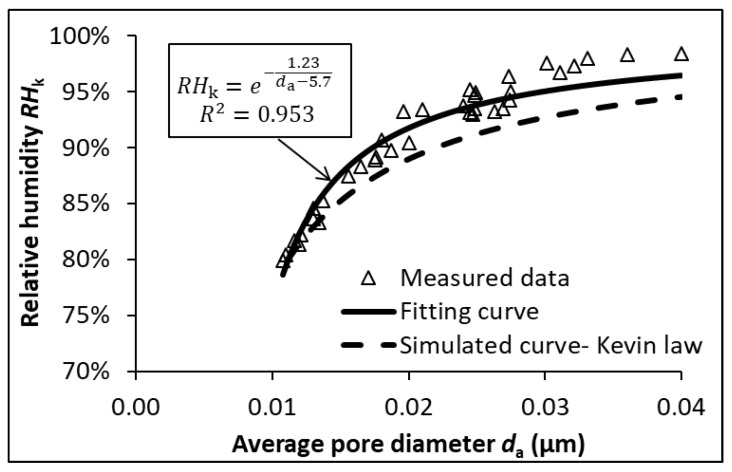
Relative humidity expressed as a function of average pore diameter.

**Figure 11 materials-12-01214-f011:**
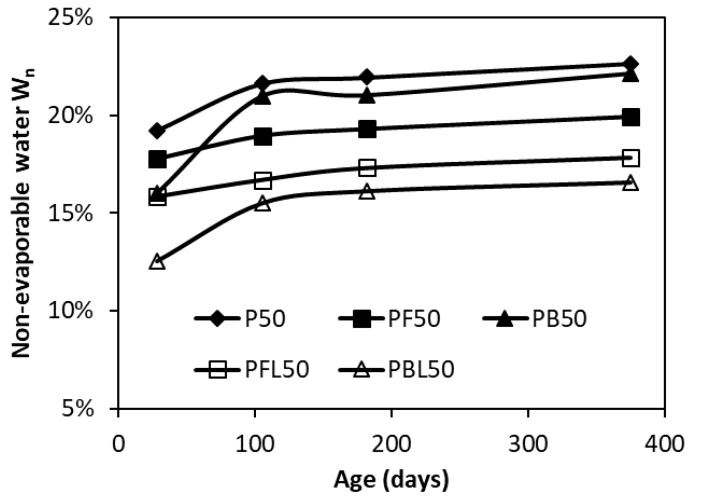
Evolution of non-evaporable water with age in various cementitious pastes.

**Figure 12 materials-12-01214-f012:**
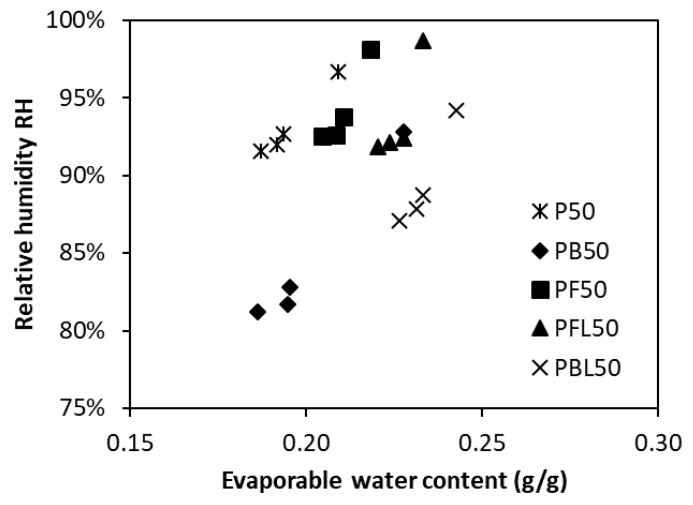
Effect of evaporable water content on relative humidity in various cementitious pastes.

**Table 1 materials-12-01214-t001:** Chemical composition of ordinary Portland cement (OPC), fly ash (FA), ground granulated blast furnace slag (BFS) and limestone powder (LP) by X-ray fluorescence (g/100 g).

Items	Raw Materials			
	OPC	FA	BFS	LP
CaO	64.495	5.537	41.398	-
SiO_2_	18.875	50.554	34.015	0.737
Al_2_O_3_	4.481	30.743	11.117	0.180
Fe_2_O_3_	3.686	6.301	0.529	0.073
MgO	2.012	1.009	8.284	0.523
K_2_O	0.508	1.109	0.398	0.026
Na_2_O	0.341	0.284	0.205	0.010
SO_3_	2.625	0.785	2.430	0.082
TiO_2_	0.319	2.362	1.027	0.020
CaCO_3_	1.185	-	-	98.316
Others	3.824	1.316	0.597	0.033
LOI%	3.04%	2.66%	0.38%	43.26%
Specific gravity (g/cm^3^)	3.12	2.26	2.87	3.08

**Table 2 materials-12-01214-t002:** Mix proportions (weight percentage) used for the binders.

Mixtures	OPC	FA	BFS	LP	Water-Binder-Ratio (w/b)
P35	100%	-	-	-	0.35
P40	100%	-	-	-	0.4
P50	100%	-	-	-	0.5
P60	100%	-	-	-	0.6
PF50	70%	30%	-	-	0.5
PFL50	65%	30%	-	5%	0.5
PB40	30%	-	70%	-	0.4
PB50	30%	-	70%	-	0.5
PB60	30%	-	70%	-	0.6
PBL50	25%	-	70%	5%	0.5

**Table 3 materials-12-01214-t003:** Parameters of Equation (11) for OPC binders.

Mixtures	Water-Cement-Ratio	A	n	RH_0_	R^2^
P35	0.35	0.154	36.87	0.82	0.9925
P40	0.4	0.118	42.47	0.86	0.9924
P50	0.5	0.075	68.75	0.91	0.9893
P60	0.6	0.044	106.76	0.94	0.9657

**Table 4 materials-12-01214-t004:** Amount of ettringite, monocarbonate (Mc) and calcium hydroxide (CH) relative to dry cement content by wt.%.

Paste Mixtures	P50	PF50	PB50	PFL50	PBL50	Density (kg/m^3^)
Ettringite	0.1%	2.3%	0.2%	4.5%	0.2%	1778 [23]
Mc	4.3%	0.2%	0.2%	8.6%	3.1%	2175 [24]
CH	20.1%	8.2%	2.3%	10.5%	1.1%	2251 [25]

**Table 5 materials-12-01214-t005:** Pore solution chemistry (mol/L) of cement pastes cured under sealed condition for 28 days.

Mixtures	By Inductively Coupled Plasma Optical Emission Spectrometry (ICP-OES)							By Calculation
	Na	K	Ca	Mg	Al	Si	Fe	OH
P35	0.272	0.478	0.0055	1.88 × 10^−4^	1.42 × 10^−3^	0.0095	1.21 × 10^−4^	0.750
P40	0.257	0.428	0.0013	<8.75 × 10^−5^	1.07 × 10^−4^	0.0011	<37.5 × 10^−5^	0.685
P50	0.197	0.27	0.0037	1.00 × 10^−4^	2.63 × 10^−4^	0.0018	6.79 × 10^−5^	0.467
P60	0.155	0.211	0.0023	4.50 × 10^−5^	4.01 × 10^−4^	0.0007	<19.6 × 10^−5^	0.366
PF50	0.172	0.181	0.0051	1.29 × 10^−4^	3.44 × 10^−4^	0.0038	9.64 × 10^−5^	0.353
PFL50	0.142	0.154	0.0006	<4.58 × 10^−5^	5.56 × 10^−5^	0.0005	<19.6 × 10^−5^	0.296
PB40	0.107	0.119	0.0009	<21.20 × 10^−5^	4.89 × 10^−4^	0.0018	<9.1 × 10^−5^	0.226
PB50	0.089	0.097	0.0048	7.71 × 10^−4^	1.29 × 10^−3^	0.0034	7.14 × 10^−5^	0.186
PB60	0.085	0.091	0.0011	<5.00 × 10^−5^	4.52 × 10^−4^	0.0007	<2.14 × 10^−5^	0.168
PBL50	0.081	0.087	0.0015	<4.6 × 10^−5^	1.19 × 10^−4^	0.0003	<1.9 × 10^−5^	0.168

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
