# Peer review of "Microstructure-Based Relative Humidity in Cementitious System Due to Self-Desiccation"

_materials, 2019, doi:10.3390/ma12081214_

Reviewer 1 Report

In submitted paper the influence between RH and microstructure is studied. Curing age of tested cement pastes is 1 day up to 1.5 year.

Cement hydration is complex process. The paper is focused mainly on formation C3A, respectively Aft, AFm. What about formation other clinker minerals?

Decreasing of porosity is explain due to pozzolanic reaction. Pozzolanic reaction really leads to finer pore structure but it must be supported by data, would not be worthwile to add XRD patterns, pore size distribution diagram?

Very important reaction affecting the microstructure is carbonation, which in neglected in this paper why?

Many references in the paper are older than 20 years.

Author Response

Reviewer 1#: In submitted paper the influence between RH and microstructure is studied. Curing age of tested cement pastes is 1 day up to 1.5 year.

Cement hydration is complex process. The paper is focused mainly on formation C3A, respectively Aft, AFm. What about formation other clinker minerals?

Response: This manuscript focuses on studies of the relationship between pore structure and relative humidity. Cement hydration is not the first important issue to be discussed in that regard. What really matters is the microstructural formation, especially the pore structure formation. The hydration of C3A and the hydrates Aft/AFm are presented in the manuscript, just for providing explanations of the synergistic effect between fly ash and limestone powder, whereby the pore structure can be densified.

Decreasing of porosity is explain due to pozzolanic reaction. Pozzolanic reaction really leads to finer pore structure but it must be supported by data, would not be worthwile to add XRD patterns, pore size distribution diagram?

Response: Thank you for the comments. Pore structure refinement due to pozzolanic reactions is a common conclusion and has been widely recognized. It is therefore reasonable for me only to give some references, like [6][8][19][26][27], to support this conclusion. This manuscript focuses on studies of microstructure (pore structure). Cement hydration and pozzolanic reactions will, of course, influence the microstructure, but they are not the emphasis in this manuscript. Otherwise, the words of this paper will be too much.

The main contribution of this paper is the proposal of the new relationship, Eq. (12), as presented in Figure 9. This relationship can be very helpful when studying the moisture distribution and related transport properties of cementitious materials. In this respect you can find another paper of mine, e.g. [4], and more research about this point will be reported in the near future.

Very important reaction affecting the microstructure is carbonation, which in neglected in this paper why?

Response: This manuscript investigates the self-desiccation and associated reduction of relative humidity. Self-desiccation is defined as a drying state that results in loss of capillary water and reduction of capillary humidity. Self-desiccation is only related to the hydration process. For the studies of self-desiccation, all samples should be cured under sealed condition. Carbonation has nothing to do with self-desiccation process, therefore not mentioned in this manuscript.

Many references in the paper are older than 20 years.

Response: According to a comprehensive literature review, previous scholars still play a dominant role in the knowledge of self-desiccation and moisture properties of cementitious materials. No new theory or breakthrough regarding cementitious moisture properties was proposed in a recent decade. The scholars, such as Snyder, Bentz, Flatt, Scherer, Bullard, Parrott, Jensen, Willis etc., still have a very high influential factor nowadays in concrete studies.

Please be noticed that, in the list of references, some are old (because they are the initiator and pioneer in the research of interest) and many others are newly published in recent years (representing the knowledge development).

Reviewer 2 Report

This is a well written well-organized draft. The draft is about the relation between the microstructure of the cementitious system and RH due to self-desiccation. However, the reviewer believes that the title can be elaborate better in order to increase the readers' interest and improve the authors' implication. These are questions to improve the readability and quality of the article:

1.  Freezing at -195 with liquid nitrogen can affect the microstructure dramatically, especially at higher RH. This effect can be different at different pore size too. It is important to assure that the freezing effect on microstructure is considered. At higher RH this effect can drastically change your conclusions.

2.  Geometry and size of the samples?

3.  page 10, line 290-292: "The relatively lower porosity increment ...". The reviewer believes that this sentence is a bit too strong. Perhaps rephrasing and adding refs can be good. 

4.  Measuring RH with these sensors is very sensitive, however pore size can affect this measurement too. Can authors make arguments on the effect of pore size on the actual measurement and accuracy of RH?

5.  The MIP cumulative intrusion and pore size distribution can be very useful. The reviewer believes it is worth to add those graphs.

6.  Line 306: ICP-OES was performed to investigate pore chemistry? If that is correct, please provide the result. 

Author Response

Reviewer 2#: This is a well written well-organized draft. The draft is about the relation between the microstructure of the cementitious system and RH due to self-desiccation. However, the reviewer believes that the title can be elaborate better in order to increase the readers' interest and improve the authors' implication. These are questions to improve the readability and quality of the article:

Response: The work presented in this manuscript belongs to kind of fundamental study, and was initiated to study the mechanisms hidden behind the relative humidity formation in cementitious materials. This research maybe related more to science, but less to engineering. This is the reason why this title was chosen.

1.  Freezing at -195 with liquid nitrogen can affect the microstructure dramatically, especially at higher RH. This effect can be different at different pore size too. It is important to assure that the freezing effect on microstructure is considered. At higher RH this effect can drastically change your conclusions.

Response: Thank you for the valuable comments. Regarding all the methods to dry cement pastes for pore structure analysis, the liquid nitrogen freeze-drying method has the minimum damage. Many studies have been performed and support this point. Please refer to the two representatives:

·         G. Ye, Experimental study and numerical simulation of the development of the microstructure and permeability of cementitious materials, PhD thesis, Delft University of Technology, 2003.

·         C. Galle, Effect of drying on cement-based materials pore structure as identified by mercury intrusion porosimetry A comparative study between oven-, vacuum-, and freeze-drying. Cement and Concrete Research 31 (2001) 1467–1477.

The influences of drying procedure on pore structure changes have been a debate for a long time. However, there is one thing we need to consider that the tensile strength of porous materials will increase significantly with decrease of the scale (mm → µm → nm) and also with the decrease of the sample size.

2.  Geometry and size of the samples?

Response: For pore structure analysis, the cubic sample size is 1 cm3 (in total 4~8 gram for one MIP test), as described already in manuscript of section 2.2.

For XRD analysis, the powder samples are smaller than 10 micros (in total 3 gram for one test), as described already in subsection 2.3.4.

3.  page 10, line 290-292: "The relatively lower porosity increment ...". The reviewer believes that this sentence is a bit too strong. Perhaps rephrasing and adding refs can be good.

Response: Thank you for the comments. A similar study previously reported on this point can support this statement. Please refer to: [K. De Weerdt, K.O. Kjellsen, E.J. Sellevold, H. Justnes (2011), Synergistic effect between fly ash and limestone powder in ternary cements, Cement and Concrete Composites 33:30–38]. This reference has been added in the revised manuscript.

4.  Measuring RH with these sensors is very sensitive, however pore size can affect this measurement too. Can authors make arguments on the effect of pore size on the actual measurement and accuracy of RH?

Response: Thank you for the comments. Two rotronic HC2-AW sensors were used for the RH measurements. The measurement range of the RH-sensor: 5.0…99.9 %RH and -40…85 ˚C. The accuracy of the RH-measurements is 0.5 %RH (at 23±5 ˚C).

Based on the knowledge of the Kelvin-Laplace effect, the curvature formed in the liquid-vapour interface determines the RH level in the capillary pore structure. The relationship between RH level and pore size can be described by the Kelvin-Cogen equation. Please refer to one previous paper of mine, see reference [12]. More principles and details can be found in [Neimark A.V., Ravikovitch P.I., Vishnyakov A. (2003) Bridging scales from molecular simulations to classical thermodynamics: density functional theory of capillary condensation in nano pores. J Phys Condens Matter 15:347-365].

The diameter of one water molecule is around 0.4 nm. If the pore size of interest is too small, the water in the pore cannot be sufficiently curved layer by layer. The applicability of the Kelvin law for RH determination is fine for pore diameter above 8 nm, please see reference [Fisher L.R. (1981) Experimental studies on the applicability of the Kelvin equation to highly curved concave menisci. J Colloid Interface Sci 80:528-541]. For pore diameter below 8 nm, the accuracy of RH measurement becomes progressively poor. Note that the pore structure obtained in this manuscript was based on the pore sizes above 7 nm.

5.  The MIP cumulative intrusion and pore size distribution can be very useful. The reviewer believes it is worth to add those graphs.

Response: I cannot agree more with your comments. Thank you. Indeed, the information of pore sizes is very useful. In this study ten mixtures with five curing age, in total 50 different samples, were prepared and their pore structures were all measured by MIP. If I add all those graphs in the manuscript, it will be too much. Therefore, I add one representative figure of the pore size distribution of cement paste, as shown in Figure 4 of the revised manuscript.

6.  Line 306: ICP-OES was performed to investigate pore chemistry? If that is correct, please provide the result.

Response: Thank you for the comments. ICP-OES has proved to be an effective measure for determining the amount of cation in the pore solution. The result of paste specimens at 28 days, as a representative, is provided in Table 5 of the revised manuscript.

Reviewer 3 Report

The article attempts to find a link between relative humidity of different cement pastes and their porosity, water-to-cement ratio and other properties. Even if the manuscript may be interesting I do not recommend it for publication due to the following reasons:

- What was the reason for the selection of range of paste mixtures? Why these binders were selected? Why some of these pastes had 30% of OPC and 70% of BFS and other replacement levels of OPC by SCMs were 30% for FA and 5% for LP by mass of the binder? Is it really enough to reliable discuss the effect of SCMs on RH (see point 3.1.2). In my opinion the methodology is not well conducted for such a statement formulated in point 3.1.2,

- I do not see significant scientific explanation of the presented relations between water-to-cement ratio calculated at the time of the mixing and the relative humidity measured after some time (see for example fig. 5),

- No details about sample curing was discussed. I only suppose that the relative humidity was measured on saturated samples while the total porosity was measured on dry samples. Is it true? If yes, how is it possible to link the relative humidity measured on saturated samples with total porosity measured on dry samples? If not, how is it possible to exist the cement paste sample with over than 95% of relative humidity and over 30% of porosity at the same time? Without the proper methodology it is not significant to state such a general equations showing negatively and exponentially correlations between relative humidity and the curing age (see equation 9),

- The cement paste samples with 40% of total porosity suggest that the samples were not properly densified and/or cured. The reviewer is not able to validate it because no Details on curing and densification was provided.

Overall too many questions are related to the article with the special emphasis on the methodology, discussion and analysis.

Author Response

 Reviewer 3#: The article attempts to find a link between relative humidity of different cement pastes and their porosity, water-to-cement ratio and other properties. Even if the manuscript may be interesting I do not recommend it for publication due to the following reasons:

- What was the reason for the selection of range of paste mixtures? Why these binders were selected? Why some of these pastes had 30% of OPC and 70% of BFS and other replacement levels of OPC by SCMs were 30% for FA and 5% for LP by mass of the binder? Is it really enough to reliable discuss the effect of SCMs on RH (see point 3.1.2). In my opinion the methodology is not well conducted for such a statement formulated in point 3.1.2,

Response: Water-to-binder ratio (w/b) and blended materials are two common factors that are widely used to alter the pore structure of cementitious systems. This research work was carried out in the Delft University of Technology, the Netherlands. All the raw materials were from the Netherlands. The replacement levels, i.e. FA 30%, BFS 70%, and LP 5%, are typically observed in practice in the Netherlands.

This manuscript intends to find a general link between pore structure and relative humidity. Utilization of w/b and blended materials is merely a method to alter the pore structure. The main contribution of this manuscript lies in the establishment of a new model, as described by Eq. (12). This model covers a range of mixtures of different w/b and blends, and in this model the concept of average pore diameter is adopted to represent the information of pore characteristics. No matter what w/b or cement type is, there should be an average pore diameter of the porous system of interest. While of course, whether this model is universally valid for all cementitious materials, further research can be done in future. However, this work provides an initiative way to correlate the pore structure (a general concept) and relative humidity due to self-desiccation. More importantly, it is new and also proved to be effective in this research by using the average pore diameter to represent the pore structure information and further to relate the capillary pore humidity. The author has utilized this model to predict the moisture related transport properties in cementitious materials, and found it is really useful, see references [4][14].

Self-desiccation can result in chemical shrinkage, autogenous shrinkage, cracking and also lead to changes of mechanical behaviour and changes of transport properties of concrete, and so on. All these provide considerable added values of this research work.

- I do not see significant scientific explanation of the presented relations between water-to-cement ratio calculated at the time of the mixing and the relative humidity measured after some time (see for example fig. 5),

Response: Thank you for the comments. The relationship shown in Fig. 5 (it is Fig. 6 in the revised manuscript) is based on regression analysis. This relationship is obtained on curve-fitting basis, it is not a model, therefore scientific explanations are unseen. However, the relationship is supported by a range of experimental data, and it is quite reasonable at least for the w/b ratios studied in this research, i.e. 0.35~0.6.

- No details about sample curing was discussed. I only suppose that the relative humidity was measured on saturated samples while the total porosity was measured on dry samples. Is it true? If yes, how is it possible to link the relative humidity measured on saturated samples with total porosity measured on dry samples? If not, how is it possible to exist the cement paste sample with over than 95% of relative humidity and over 30% of porosity at the same time? Without the proper methodology it is not significant to state such a general equations showing negatively and exponentially correlations between relative humidity and the curing age (see equation 9),

Response: Please understand that this research work investigates the relative humidity RH caused by self-desiccation. Self-desiccation means unhydrated cement in paste or concrete continues to hydrate accompanying the consumption of free water and consequently results in RH reduction. The concept of self-desiccation, in itself, means that all the samples were cured under sealed condition, which has already been described in the experimental program, see section 2.2. Saturated samples cannot be used for RH measurements, since the RH can be approaching 100%.

In this research work,

·         First, the RH sensors were used to monitor the changes of relative humidity due to continuous cement hydration.

·         Second, parallel cement paste samples with the same mixtures were cured under sealed condition. Sealed condition means that the pore water will be progressively consumed with cement hydration and the RH will decrease accordingly. After 28, 105, 182, 370 and 575 days, all samples were undergone freeze-drying procedures and then used for pore structure measurements by MIP technique.

All these experimental works are widely adopted by the scholars in the research field of cement and concrete, and already proved to be reasonable and reliable.

- The cement paste samples with 40% of total porosity suggest that the samples were not properly densified and/or cured. The reviewer is not able to validate it because no Details on curing and densification was provided.

Response: All the samples were under standard curing, well prepared, and have been used in many of previous published papers/book, for example References [3][4][12][14].

The porosity values were obtained by MIP measurements. The porosity data are comparable to those of other researchers, such as [Yu Z.Q, Ye G. The pore structure of cement paste blended with fly ash. Constr Build Mater 2013(45):30-35.].

As a matter of fact, the porosity of FA-blended pastes should be higher than that of the ordinary Portland cement (OPC) pastes, one reason is due to the relatively slow chemical reactions of FA and another reason lies in porous FA particles themselves. FA particles are hollow in nature. These hollow voids can be filled with mercury in MIP measurements, resulting in higher porosity than expected. When the FA particles are incorporated in cementitious systems, the total porosity (including the hollow voids of FA) can be much higher than that of OPC paste, especially at the curing age of 28 days (w/b=0.5). Moreover, the addition of limestone will dilute the cementitious systems, further increasing the total porosity.

In Fig. 8, the highest total porosity can be ascribed to the mixture PFL50, which contains 65%OPC + 30%FA + 5%LP. At 28 days, its porosity is around 36%. This is quite normal, common and reasonable, which has been reported in many of previous reports. While after 1 year, its total porosity drops to nearly 30%. For the majority of the mixtures used in this study, the total porosity after 1 year is below 22%.

It is not reliable to judge densification merely based on porosity, because porosity is influenced by many factors such as w/b ratio and curing age and blends etc.

Round  2

Reviewer 1 Report

The paper may be published as it is.

Reviewer 3 Report

The answers are satisfied. Thus, I ommend this paper for publication.